# Time Course RNA-seq Reveals Soybean Responses against Root-Lesion Nematode and Resistance Players

**DOI:** 10.3390/plants11212983

**Published:** 2022-11-04

**Authors:** Valéria Stefania Lopes-Caitar, Rafael Bruno Guayato Nomura, Suellen Mika Hishinuma-Silva, Mayra Costa da Cruz Gallo de Carvalho, Ricardo Vilela Abdelnoor, Waldir Pereira Dias, Francismar Corrêa Marcelino-Guimarães

**Affiliations:** 1Department of Biological Sciences, Universidade Estadual de Londrina (UEL), Londrina 86057-970, PR, Brazil; 2Brazilian Agricultural Research Corporation-Embrapa Soja, Londrina 86001-970, PR, Brazil; 3Department Biochemistry and Biotechnology, Universidade Estadual de Londrina (UEL), Londrina 86057-970, PR, Brazil; 4Department of Biological Sciences, Universidade Estadual do Norte do Paraná (UENP), Bandeirantes 86360-000, PR, Brazil

**Keywords:** migratory nematode, molecular basis of host defense, resistance, transcriptome in different perspectives, soybean reference genes in *P. brachyurus* infection

## Abstract

*Pratylenchus brachyurus* causes serious damage to soybean production and other crops worldwide. Plant molecular responses to RLN infection remain largely unknown and no resistance genes have been identified in soybean. In this study, we analyzed molecular responses to RLN infection in moderately resistant BRSGO (Chapadões—BRS) and susceptible TMG115 RR (TMG) *Glycine max* genotypes. Differential expression analysis revealed two stages of response to RLN infection and a set of differentially expressed genes (DEGs) in the first stage suggested a pattern-triggered immunity (PTI) in both genotypes. The divergent time-point of DEGs between genotypes was observed four days post-infection, which included the activation of mitogen-activated protein kinase (MAPK) and plant–pathogen interaction genes in the BRS, suggesting the occurrence of an effector-triggered immunity response (ETI) in BRS. The co-expression analyses combined with single nucleotide polymorphism (SNP) uncovered a key element, a transcription factor phytochrome-interacting factor (PIF7) that is a potential regulator of moderate resistance to RLN infection. Two genes for resistance-related leucine-rich repeat (LRR) proteins were found as BRS-specific expressed genes. In addition, alternative splicing analysis revealed an intron retention in a myo-inositol oxygenase (MIOX) transcript, a gene related to susceptibility, may cause a loss of function in BRS.

## 1. Introduction

Plant-parasitic nematodes (PPN) are an important agriculture pathogen, causing yield losses estimated from USD 80 billion to USD 157 billion per year worldwide [1]. Until recently, the *Meloydogyne* spp. genera was considered the most important nematode threat to major crops in the United States, South America, and South Asia, followed by *Heterodera* and *Globodera* species [2]. However, root-lesion nematode (RLN; *Pratylenchus* species) has recently risen in prominence, ranking as the second most important phytopathogenic nematode in Brazil, the world largest soybean producer [3,4]. *Pratylenchus brachyurus* has caused over 30% soybean crop loss in Brazil [5]. RLN also has been reported in the southeastern and central United States, South America, Africa, South Asia, Western Australia, and Europe [6,7]. RLN is a migratory root endoparasite with a wide host range, which is difficult to manage. *Pratylenchus brachyurus* females reproduce by parthenogenesis, and have a life cycle of ~45 days, depending on the environmental conditions [8]. Eggs hatch approximately one week after deposition, and all four juvenile and adult stages can infect and feed on plant roots [9]. Unlike sedentary nematodes, RLN infection does not produce a feeding site or host cell differentiation, instead, results in the development of necrotic spots or lesions due to nematode movement and feeding in the plant root cortex. *Pratylenchus sp.* host resistance in wheat genotypes suppresses migration, juvenile maturation, and reproduction rather than infection or penetration [10].

Most of the plant resistance genes identified to date are related to sedentary nematodes, probably due to their biotrophic behavior and intimate relationship with the host [8,11]. For example, the *Mi* gene in tomato and *N* gene in pepper are related to the root-knot nematode (RKN—*Meloidogyne* spp.) resistance [12]. Additionally, the *Gro*1-4 gene was described as a class of toll interleukin 1 receptor-nucleotide binding leucine-rich repeat (TIR-NB-LRR or TNL) proteins in tomato capable of conferring resistance to *Globodera rostochiensis* (pathotype *Ro*1) [13]. *Rhg1* is a soybean gene that regulates stress and defense genes against *Heterodera glycines* (SCN) [14]. The stone fruit trees, *Prunus* spp., has genes that confer resistance to *Meloidogyne incognita* and *Meloidogyne arenaria*, such as *Ma* and *Rjap* (*Prunus cerasifera*—plums), *RMia* (*Prunus persica*—peach), and *RMja* (*Prunus dulcis*—almonds) [15,16]. *Ma* genes are members of the TIR-NB-LRR gene family, which completely prevents nematode proliferation and gall formation [17]. Other resistance genes include *Gpa*2, which belongs to the leucine-zipper nucleotide-binding site leucine-rich repeat (LZ-NBS-LRR)-containing class of R genes from *Solanum tuberosum* resistant to potato cyst nematode (PCN), *Globodera pallida*; *rkn*1 from *Gossypium hirsutum* resistant to *M. incognita*, and *Hs*1pro−1 from *Beta procumbens* resistant to the beet cyst nematode, *H. schachtii* [18,19,20,21].

Few soybean genotypes show resistance and tolerance to RLN, however wheat mapping populations have revealed QTLs for resistance. [22,23]. A single gene conferring resistance to *P. neglectus* and *P. thornei* has been mapped in wheat chromosome 7AL [24], and a major resistance quantitative trait locus (QTL) was identified on chromosome 6DS [25]. Based on the screened-out barley population, five major QTLs (*Pne*3H-1, *Pne*3H-2, *Pne*5H, *Pne*6H, and *Pne*7H) mapped on four linkage groups (3H, 5H, 6H, and 7H) were associated with *P. neglectus* resistance [26]. A highly significant QTL, *QPnToIMI.1*, was determined to be related to resistance to *Pratylenchus neglectus* in the legume *Medicago littoralis* [27].

Plants have complex defense mechanisms against pathogen attack, involving structural and chemical barriers as well as induction of defense-related genes, such as pathogen-related proteins (PR proteins) [28]. PR proteins are a component of Pathogen-Associated Molecular Pattern (PAMP)-triggered immunity (PTI), the first defense line in plants. This defense mechanism either signals for systemic acquired resistance (SAR); or it can directly fight against pathogenic invasion. However, if a pathogen can disrupt host PTI with an effector molecule, resistant plants (present R genes) can initiate a defense response by ETI (Effector-Triggered Immunity) [29]. The plant immune system is a “zigzag” model of plant–pathogen interactions [30].

Genome-wide approaches can help explain plant–pathogen interactions, including transcriptomes to provides insights into host defense responses using gene expression and alternative splicing. To date, there are no transcriptome studies on soybean response to migratory nematodes, and few RNA-seq analyses of responses in other plant species. Rice response against the migratory root rot nematode (RRN), *Hirschmanniella oryzae*, showed the induction of programmed cell death and oxidative stress in addition to the obstruction of the normal metabolic activity of the root [31]. Furthermore, RNA-seq analysis in rice upon root knot (*Meloidogyne graminicola*) and root rot nematode (RRN, *Hirschmanniella oryzae*) infection at two time-points (3 and 7 days after infection—dai), showed hormonal signaling pathways, such as jasmonate (JA) and ethylene (ET) induction [32]. *Boehmeria nivea* (L.), ramie, showed 137 significantly differentially expressed genes, such as those for protease inhibitors, pathogenesis-related proteins (PRs), cell wall reinforcement, and transcription factors (TF) in response to *P. coffeae* infection [33]. Multi-layered defense mechanisms were induced in a transcriptome analysis of oak trees against *P. penetrans*, comprising reactive oxygen species formation, hormone signaling (e.g., jasmonic acid synthesis), and proteins involved in the shikimate pathway [34].

This is the first publicly available transcriptome of the interaction between RLN and soybean. Thus, our initial objective was to explore and compare genes expressed in these contrasting genotypes over the course of RLN infection. We were able to identify metabolic pathways and processes that are modulated by infection and to observe the expression of PTI-related genes in the susceptible genotype while in the moderately resistant genotype the expression profile was closer related to the ETI response, endorsing our hypothesis. Additionally, by combining co-expression of DEGs and SNP analysis we identified key genes and/or important signaling pathways involved in soybean response to RLNs. Our findings may help develop crop technologies to mitigate migratory nematodes.

## 2. Results

### 2.1. DEGs Reveal Two Main Stages of Gene Regulation in Response to RLN

To reveal the transcriptional response associated with RLN infection in moderately resistant (BRS) and susceptible soybean genotypes (TMG), both RLN-infected and non-infected samples were subjected to RNA-seq analysis at four different time-points. The transcriptome depth and coverage were accessed with rarefaction analysis, showing that all 16 libraries reached the plateau of mapped genes with the obtained number of reads (Appendix A).

The principal component analysis (PCA) of the normalized transcriptomic data revealed at first glance that the genotypes have distinct gene expression profiles at different time-points. Importantly, the variation between RLN inoculated and non-inoculated samples represent the main factor of differences in gene expression (Figure 1a).

We found a total of 900 unique DEGs during the soybean response to RLN, 425 (401 found at only one time-point + 24 found at two or more time-points) exclusive genes in BRS, 355 (342 found at only one time-point + 13 found in two or more timepoints) exclusive genes in TMG; and 120 co-express common differentially expressed genes in both genotypes (Figure 1b). Additionally, hierarchical clustering results showed that the 157 DEGs (24 DEGs were found at two or more time-points only in BRS, 13 DEGs at two or more time-points only in TMG, and 120 common DEGs between genotypes) (Figure 1b and Appendix A) are mostly co-expressed, presenting similar expression profile patterns among time-points in the genotype or between genotypes [35], which gene expression induction was more evident than repression (Figure 1c, Appendix A).

The total number of DEGs was higher at 8 dpi in both genotypes (316 DEGs in BRS and 200 in TMG, Figure 1c and Appendix A). However, at this time-point, while BRS showed a general up-regulated profile, TMG DEGs were down-regulated. Interestingly, the opposite general expression profile was observed at 2 dpi, when the average fold-change was down-regulated in BRS while up-regulated in TMG (Figure 1c). The lowest number of DEGs was observed at 4 dpi for both genotypes, being 70 DEGs for BRS and 97 for TMG. Thus, although genotypes had distinct expression profiles (Figure 1a), overall, changes in the soybean transcriptome showed similar points of regulation with an initial response at 1 and 2 dpi, and another regulation peak at 8 dpi (Figure 1c,d). The time-point 4 dpi, despite presenting the lowest number of DEGs in both genotypes, was the moment when genotype response to RLN infection was most differentiated (Figure 1d). At 4 dpi, TMG maintained a response closer to those observed at 1 and 2 dpi, contrasting to BRS that at 4 dpi was clearly close to the responses observed at 8 dpi (Figure 1d).

### 2.2. RLN Infection Redirects Soybean Metabolism

Secondary metabolites biosynthesis, metabolic pathways, and phenylpropanoid biosynthesis were enriched in the 120 co-expressed genes, BRS 425 DEGs and TMG 355 DEGs (Figure 2a—cluster 1, Appendix A), where the key pathway genes include: Glyma.04G121700 and Glyma.15G156100 significantly up-regulated in both genotypes and at all time-points in the secondary metabolite biosynthesis; and Glyma.12G054200 up-regulated in both genotypes in phenylpropanoid biosynthesis. Notably, many other phenylpropanoids-related pathways were found to be enriched in response to RLN. Flavone and flavonol biosynthesis were found to be enriched only in BRS, including Glyma.05G021900 down-regulated (Figure 2a—cluster 2, Appendix A). Flavonoid biosynthesis was enriched in both genotypes, but not among the co-expressed genes. The most relevant genes related to flavonoid biosynthesis in BRS were Glyma.08G312000, Glyma.05G223400, Glyma.05G021900, and Glyma.18G113100 down-regulated and Glyma.19G105100 and Glyma.06G202300 up-regulated. Isoflavonoid biosynthesis was enriched with a 0.10 ratio only in the co-expressed genes, including Glyma.09G048900, Glyma.15G156100, Glyma.09G049200, Glyma.09G049300, and Glyma.18G080400 all of which were significantly up-regulated (Figure 2a—cluster 2, Appendix A). In addition, BRS had an activation on stilbenoid, diarylheptanoid, and gingerol biosynthesis, not observed in TMG (Figure 2a—cluster 2, Appendix A). MAPK signaling was enriched in both genotypes, but with emphasis in BRS for Glyma.03G162500, Glyma.03G162700, and Glyma.10G007000 (ethylene-responsive transcription factor 1 (ERF1)), and Glyma.09G255000 (1-aminocyclopropane-1-carboxylate synthase-like protein 1), all four of which were up-regulated at 8 dpi, along with two genes classified by MAPK–plant–pathogen interaction, Glyma.02G006200 (ethylene-responsive transcription factor 1 (ERF1)) and Glyma.03G088800 (cGMP-dependent and protein kinase C) (Figure 2a—cluster 2). On the other hand, carbon fixation and glyoxylate and dicarboxylate metabolic pathways were boosted only in TMG co-expressed genes. (Figure 2a—cluster 4).

Zeatin biosynthesis was found to be significantly enriched in BRS, only at 8 dpi, with three important pathway genes up-regulated: Glyma.09G225400, Glyma.02G184200, and Glyma.10G107900 (Appendix A). Genes related to glyceollins (phytoalexins) biosynthesis, Glyma.01G134600, Glyma.20G245100, Glyma.10G295300, Glyma.08G274800, and Glyma.13G152814 were induced at least one time-point in both genotypes; however, in BRS at 8 dpi, this induction was found to be more important for 4 of these genes (Appendix A).

Based on DEGs, the first set of responses was related to oxidative stress, calcium ion binding, and cell wall (Figure 2b). At 1 dpi, oxidative stress-related genes were up-regulated in BRS and TMG, but only in BRS at 8 dpi (Glyma.11G051800, Glyma.11G062600, Glyma.09G049200, and Glyma.13G285300). In contrast, calcium ion binding encoding genes were regulated only in BRS but showed down-regulation at 1 dpi (Glyma.10G084000, Glyma.02G182900, Glyma.02G192700, Glyma.10G087100, and Glyma.14G035800). Molecular function and cell wall-related genes were repressed in BRS at 2 dpi but not in TMG. (Glyma.01G146000, Glyma.06G314100, Glyma.09G193500, Glyma.13G186100, Glyma.15G223800, and Glyma.10G150600). The second peak of transcript modulation, at 8 dpi, genes encoding photosynthesis, protein complexes, and membrane parts were repressed only in BRS, highlighting Glyma.08G304200, Glyma.01G174400, Glyma.04G112800, Glyma.06G321900, Glyma.08G204800, Glyma.09G087700, Glyma.09G250800, Glyma.10G032200, Glyma.13G046700, Glyma.13G127200, Glyma.19G045800, Glyma.18G114900, and Glyma.18G018900. Instead, TMG showed a repression for genes related to localization establishment of and transport at 8 dpi, including Glyma.02G224600, Glyma.04G220700, Glyma.07G113100, Glyma.08G033200, Glyma.08G037200, Glyma.08G120100, Glyma.09G276800, Glyma.11G066000, Glyma.11G066300, Glyma.11G223900, Glyma.11G238500, and Glyma.14G126500.

### 2.3. Co-Expressed Gene Analysis Reveals a Network of Photosynthesis Related Genes Differentially Regulated between Genotypes

Among the 157 DEGs co-expressed, three of these genes were found as significantly induced at all time-points in TMG and BRS: Glyma.15G156100, encoding for an isoflavone 2’-hydroxylase; Glyma.15G203500, encoding for a cytochrome P450 CYP2 subfamily; and Glyma.04G121700, encoding for a catechol oxidase/tyrosinase (Appendix A). Four defense-related cysteine-rich secretory proteins containing SCP domain, in a tandem location on chromosome 13 (Glyma.13G251600, Glyma.13G251700, Glyma.13G252000, and Glyma.13G252400), were induced at 1, 2, and 8 dpi in both genotypes. Four luteolin triglucuronide degradation peroxidases (Glyma.06G145300, Glyma.09G277800, Glyma.09G277900, and Glyma.20G001400) were induced at 1 dpi in both genotypes. In addition, three chitinase-related genes (Glyma.11G124500, Glyma.03G247500, and Glyma.20G225200) were induced at 1 and/or 8 dpi in both genotypes (Appendix A).

In contrast, 18 co-expressed genes, mainly associated with the photosynthesis process, showed a significantly inverted expression profile between genotypes, nine of these at the same time-points (Figure 3a, Appendix A).

At 4 dpi, photosynthesis-related genes were repressed in TMG and induced in BRS (Figure 3a). Thus, based on the observation that these 18 genes are co-expressed in response to RLN infection in BRS and TMG, we verified whether these genes show the same pattern in other situations in soybean and/or other plants. Co-expression data from the String Database showed that none of the 18 genes have been previously found with this co-regulation in soybean (Figure 3b), but 13 out of these were found to be highly co-regulated in *Arabidopsis thaliana* and *Oryza sativa* (Figure 3c). Furthermore, String analyses identified that these genes encode proteins in an interaction network (Figure 3d).

Promoter analysis of these 18 genes identified a motif sequence that presents high similarity with a non-canonical E-box *cis* element (CANNTG), known to be targeted by *bHLH* transcription factors (Figure 3e). Comparative analysis of the *Arabidopsis* motif database indicates that this *cis* element is likely to be a binding site for the AT5G61270 gene, annotated as transcription factor PIF7 and sterol regulatory element-binding protein (SREBP).

### 2.4. Two BRS-Specific Expressed Genes Are Toll—Interleukin 1—Resistance/LRR-NB-ARC Domain

Among the 43,481 annotated soybean genes (Phytozome V13) our whole soybean RNA-seq data mapped an average of 38,075 genes per library. Based on TPM values (>0.1), BRS and TMG showed 41,757 and 41,441 mapped genes, respectively (Appendix A). To identify genotype-specific expressed genes, we performed a data examination method called genotype-specific expressed genes (GSEGs) (see methods). This analysis identified 24 GSEGs, comparing all BRS and TMG libraries (Table 1). TMG presented 10 GSEGs that were found expressed in all TMG libraries but in no BRS libraries; whereas 14 genes were found to be GSEGs in BRS, with no expression detected in TMG. Three GSEGs in TMG located in chromosome Gm06 were in tandem.

Interestingly, 13 of the 24 GSEGs identified have no previously reported inferred functional annotation. For those presenting functional annotation, TMG showed three GSEGs located in tandem on chromosome Gm06. These three genes encode ankyrin repeat protein family related. Both genotypes have GSEGs related to oxireduction activity that could work on defense response. However, BRS have two GSEGs that encode leucine-rich repeat-containing proteins (LRRs) (Glyma.01G046900 and Glyma.03G047900) and a zinc-finger domain protein (Glyma.02G090200), which are domains usually present in plant receptors or associated with defense signaling transduction.

### 2.5. RLN Infection Induces Differential Splicing

Based on our transcriptome data, we investigated whether RLN infection could cause differential alternative splicing in each genotype. The defense response to RLN showed 605 significant differential splicing events (DSEs) in both genotypes. BRS showed 312, seven of which were found at more than one time-point. TMG showed 345 DSEs with 12 of these found at more than one time-point. The genotypes shared 52 DSEs, when considering all timepoints (Figure 4a and Appendix A). Five different types of DSE were observed in the transcripts, with events in 3′untranslated regions (A3SS) and skipping exons (SE) being the most frequently found (Figure 4b).

Comparison of skip/inclusion level differences revealed significant DSEs that were contrasting between genotypes. Glyma.05G224500, which encodes an inositol oxygenase (MIOX), showed a significant alternative splicing for positive intron retention at 1 and 4 dpi in BRS, while in TMG at 8 dpi, there was significant negative intron retention (i.e., intron is removed) (Figure 4c). The intron retention event identified in Glyma.05G224500 transcript directly impacts the protein translation by the inserting a stop codon right over the protein domain, which most likely causes its loss of function. A DSE A3SS type was identified in Glyma.04G100600 transcript significantly included at 4 and 8 dpi in BRS and skipped at 2 dpi in TMG, which could interfere in mRNA behavior, such as translation activity, stability, and subcellular localization (Figure 4d). Glyma.15G160400 encodes an anankyrin repeat protein family related and showed transcripts with the SE type of event induced at 4 and 8 dpi in BRS and repressed at 8 dpi in TMG, which completely changes the resultant protein identity (Figure 4e). Seven other DSEs were also identified to be inversely regulated between genotypes at the same time-points (Figure 3f). Among these DSEs, there are genes that encode DNA excision repair protein ERCC-5 (ERCC5, XPG, RAD2) (Glyma.15G160400), Myb-like DNA-binding domain/SWI/SNF complex-related (Glyma.19G260900), macrophage migration inhibitory factor (MIF)/Phenylpyruvate tautomerase (Glyma.07G156500), S-adenosyl-L-methionine-dependent methyltransferase/hepato cellular carcinoma-associated antigen (Glyma.11G194000), and IQ calmodulin-binding motif (Glyma.03G178200), known to related to plant defense.

### 2.6. SNPs Identified in Transcripts Region May Interfere in Important Gene Networks during RLN Infection

Based on studies that applied RNA-seq data for SNP detection [37,38,39], a variant calling analysis was carried out to access SNPs present in the transcribed regions of the genotypes, using *Glycine max* Williams 82.a4.v1 as reference genome. Together, BRS and TMG presented a total of 32,755 sites of polymorphism in comparison to the reference genome; due to the nature of our data, most variants were identified in genes and flanking areas, with few variants identified over intergenic regions.

After filtering heterozygous and alternative homozygous variants, present only in BRS in comparison to TMG and the reference genome (Williams 82), we identified 14,460 polymorphisms “exclusive” to the moderately resistant genotype (Appendix A). A total of 4476 genes showed variants in BRS, including in intergenic regions, that were not found in TMG or Williams82, both susceptible to RLN. Variants in BRS were distributed mostly in downstream (23.84%) and upstream gene regions (16.17%), and in gene coding sequences (CDS) with synonymous variants (17.63%) and non-synonymous or missense variant (16.65%) (Appendix A).

Based on impact 4173 and 164 variants were classified as moderate and high impact on BRS, respectively. Moderate impact variants, including non-synonymous and in-frame insertions or deletions; and high impact variants include frameshifts, stop gain, or losses, and splicing donors or acceptors. Soybean chromosomes Gm03, Gm04, Gm08, Gm13, and Gm14 present the highest numbers of variations, classified as high impact (Appendix A).

Among the genes identified in this variant analysis, some of them were also found in one or more analyses (DEG, GSEGs, and/or DSEs), which supports the idea that these genes play an important role in the immune response to RLN infection. More specifically, these genes are represented by: (1) Glyma.15G156100, that encodes an isoflavone 2’-hydroxylase, and (2) Glyma.02G088700, that encodes a Kinase Protein, were both co-expressed at all time-points and presented a synonymous variant (Appendix A). (3) Glyma.09G049200 and (4) Glyma.18G080400, that encode an isoflavone 2’-hydroxylase and a flavonoid 6-hydroxylase-1, respectively, showed induction under RLN infection in both genotypes and presented a missense variation at the amino acid (aa) Cys413Arg and Asp459Try (Appendix A). (5) Glyma.09G064200, showed a missense variant at the aa Phe513Leu and, as ortholog of AT5G61270, encodes a transcription phytochrome-interacting factor (PIF7)—basic helix-loop-helix (bHLH) DNA-binding protein. Interestingly, this gene presents *bHLH* domain, the same domain identified in the promoter motif analysis with our co-expressed genes (Figure 3e). Another gene, (6) Glyma.13G123000, presented a missense variant and encodes an ethylene responsive transcription factor—AP2. (7) Glyma.15G08G9100, with a missense mutation in BRS at the amino acid Asn344Ser, encode a GDSL-like Lipase/Acylhydrolase domain protein and was, overall, down-regulated in both genotypes (Appendix A). Protein tyrosine kinase, (8) Glyma.13G201400, also showed general down-regulation in both genotypes and a mutation in BRS that impacts with a frameshift at the aa Asn7.

### 2.7. Nematode Penetration Activity Is Different at Late Stages

Nematode first penetration and infection progression was analyzed by counting the number of individuals extracted from inoculated root at seven different time-points, ½, 1, 2, 4, 8, 17, and 35 days post inoculation (dpi). Both genotypes showed a similar number of penetrated individuals, between 10 and 16% of total inoculated at ½ dpi, with no statistical significance in a comparison between genotypes (Figure 5a).

A progressive increase in penetration was noticed at the following time-points: 1, 2, 4, 8 dpi, and 17 dpi, however, with no statistical significance. At 35 dpi, there was a significant difference in the number of nematodes inside the roots of TMG and BRS plants. For TMG at 35 dpi, the average number of nematodes inside the roots was more than double the total number initially inoculated.

Supporting these data, the number of reads unmapped against the soybean genome that were mapped against *Pratylenchus penetrans* transcriptome presented in general higher percentages in inoculated samples in comparison to non-inoculated samples. Additionally, percentages were higher for TMG genotype than for BRS (Figure 5b). Furthermore, the number of *P. penetrans* reads was already high at 1 dpi.

### 2.8. Serine/Arginine Rich Splicing Factor Is the Most Stable Expressed Soybean Gene under RLN Infection

We performed a gene expression coefficient of variation analysis to identify the most stable soybean genes under RLN infection. A total of 606 genes had stable expression in soybean, according to the cut-off filters (Appendix A). A serine/arginine rich splicing factor (Glyma.03G175400) was identified with the most stable expression under the conditions of our study. The top-most stable genes showed TPM values from 34 to 158 and a coefficient of variation <0.06 (Appendix A).

We evaluated the stability of the most commonly used soybean reference genes in RT-qPCR analysis, EF1-α, EF1-β, TUA, TUB, and β-actin. The geometric mean of classification by ReFinder, based on the CT values, ranked TUB and TUA as the most stable genes (Appendix A). According to RNA-seq data, these five genes had a CV between 0.14 and 0.30, which is very stable (Appendix A).

To validate our RNA-seq data, a correlation between RNA-seq expression and RT-qPCR data of four genes was evaluated. The results showed the same expression profile between RNA-seq and RT-qPCR. Furthermore, the high coefficients of correlation for Glyma.13G267500 (*R*^2^ = 0.74), Glyma.13G113100 (*R*^2^ = 0.82), and Glyma.03G044900 (*R*^2^ = 0.54) demonstrates the data reliability (Appendix A).

## 3. Discussion

### 3.1. BRS Could Have a Molecular Mechanism to Reduce RLN Reproduction

Different plant genotypes can respond differently to pathogen infection at the genome and transcriptome levels, with genomic polymorphisms and differential gene expression and splicing [40,41,42]. This study used RNA-seq data from susceptible and moderately resistant to RLN soybean genotypes infected at different time-points to investigate general responses against the nematode infection, such as differences in gene modulations between genotypes.

Overall, RLN infection changed in the expression profile of 900 soybean genes, with 120 DEGs common in both genotypes. Plans infected with migratory nematodes often have low DEG numbers. A study of RLN-infected ramie plants found 137 DEGs between control and treated libraries [33]. Similarly, low number of DEGs was reported in rice roots infected with the root rot nematode, *Hirschmanniella oryzae*, and in two different genotypes of alfalfa infected with RLN, *Pratylenchus penetrans* [31,43]. A similar ratio of up- and down-regulated DEGs follows the counts and indicates a limited scale of affected host pathways and a well-coordinated, balanced host response [43]. Furthermore, observing DEGs over time after infection reveals a biphasic response with peaks at 1–2 dpi and 8 dpi, although the latter peak is more pronounced in both genotypes. Biphasic plant defense against pathogens is common, but not with nematodes [44,45]. A transcriptome study of *Persea americana-Phytophthora cinnamomic* incompatible interaction demonstrated that genes related to reactive oxygen species (ROS), Ca^2+^ signaling, and salicylic acid pathways were activated at the early time-point (18 h post-infection), while JA signaling was absent; followed by JA activation at a late time-point of interaction (24 h post-infection) [46].

Among genes co-expressed between genotypes at 1 and 8 dpi, we found three chitinase-related genes (Glyma.11G124500, Glyma.03G247500, and Glyma.20G225200) (Appendix A). Chitinases have been reported in plant responses to nematode infection, and the hypothesis is that they target nematode eggs, as their eggshell presents a middle chitinous layer composed of a protein matrix embedded with chitin microfibrils [47]. Here, the number of nematodes in roots at 35 dpi was higher from the initial inoculum in both genotypes, but significantly higher in TMG compared to BRS. This indicates that RLN has started a new life cycle, and BRS exhibited a mechanism to limit its reproduction, possibly involving chitinase-associated genes. In support of this hypothesis, overexpression of a fungal chitinase (*Pj*CHI-1) in tomato, under the control of a synthetic promoter, pMSPOA, had negative effects on *M. incognita* reproduction [48]. In addition RLN, *Pratylenchus penetrans*, seems to induce chitinase encoding genes in alfalfa (*Medicago sativa*) [49].

At 1 and 2 dpi, oxidative stress genes and energy modulation suggests nematode migration has damaged plant tissue and caused root lesions. Activation of GO categories related to oxidoreduction reactions and calcium binding in BRS and TMG at 1 dpi suggests pathogen perception and immune response induction in moderately resistant and susceptible genotypes. [42,50,51]. Reactivation of oxidative burst genes in BRS at 8 dpi may be due to RLN biology. RLN infection assays present eggs to adults, unlike RKN and other sedentary nematodes. By 8 dpi, the inoculated adult infective forms have likely laid eggs, hatched, and begun infecting. TMG, which showed signs of susceptibility after the first wave, has no response to the second, while BRS does. Another important response observed at 8 dpi in BRS was the down-regulation of genes related to the photosynthesis and components and membrane components.

In BRS, late-stage RLN infection activates genes related to the most common plant cytokinin (CK), zeatin biosynthesis. This phytohormone increases tobacco’s resistance to *Pseudomonas syringae*, modulating plant immunity and fitness. [52]. A genome-wide association study identified allelic variation at several loci related to cytokinin biosynthesis and showed that cytokinin signaling contributes to early *Arabidopsis* immunity responses against *Ralstonia solanacearum* [53].

It is unusual to discuss photosynthesis related genes in roots, since this is not where this pathway works. However, depleting this process in BRS at 8 dpi could deprive the pathogen of a carbon source. The carbon fixation pathway is enriched only in co-expressed genes and has twice the value of TMG, another result that supports this hypothesis (Figure 2—cluster 4). The relocation of nitrogen away from fungal or bacterial infection sites has been described, and this is commonly interpreted as an attempt by the host to deprive the pathogen of essential nutrients [54,55,56]. Photosynthesis depletion could save or redirect energy by reducing photosynthesis-related gene expression and protein synthesis with a non-essential pathway. Photosynthetic suppression may be key to BRS resistance. Photosynthesis genes are the dominate component of the co-expression network, supporting this theory. BRS inhibits photosynthesis, but TMG up-regulates it. The DEGs in TMG showed general maintenance of the gene expression profile from 1 to 4 dpi, with the reorganization of DEGs at 8 dpi, while for BRS, the expression profile of DEGs at 4 dpi was maintained at 8 dpi. Thus, at 4 dpi, genotypes showed the most DEG composition differences. This difference was conserved after 4 dpi, suggesting a resistance gene expression profile is probably established at this point and corroborate the soybean response to RLN has early and late stages. At 4 dpi, a moderate resistance response redirects transcription activity, negatively affecting photosynthesis, cell wall, and membrane synthesis.

### 3.2. Metabolic Pathways Affected under RLN Infection Suggest PTI and ETI in Moderate Resistance Response

Flavonoids, terpenoids, stilbenoid, diarylheptanoid, and gingerol were among the DEG-enriched pathways related to plant defense in BRS [57,58,59,60]. Terpene, phenolic, and nitrogen compounds are used by plants as defense agents against biotic stresses, and their regulation is seen in moderately resistance response [61]. Glyceollins, major isoflavone phytoalexins in soybeans, are well-known for their antimicrobial properties [62]. Fatty acid, lipid, and flavonoid classes of metabolites in resistant wheat roots may inhibit *P. thornei* reproduction and genes related to these pathways are resistance candidates [63,64]. Lignin is a highly structured polymer of phenylpropanoid molecules that plays a role in plant resistance [65]. Lignified walls prevent pathogen infections by acting as a non-degradable mechanical barrier. The regulation of secondary metabolism genes in BRS may be a response to nematode migration and necrosis, as seen in plant–herbivore interactions [66,67].

BRS-enriched DEGs included MAPK signaling, plant hormone signal transduction, and plant–pathogen interaction genes, common plant defense players, suggesting PTI and ETI activation [68,69]. Both PTI and ETI induce a common set of downstream defense responses, including ROS, calcium influx, kinase activation, and global transcriptional reprogramming for defense. Ca^2+^ binding and N-terminal phosphorylation of conserved residues change oxidases’ conformation to generate ROS. These PTI and ETI mechanisms use different kinases [70]. Downstream MAPK cascades are also induced similarly, but it is unknown whether the upstream activating kinases for PTI and ETI are identical [71]. Thus, different kinases may converge at critical signaling nodes with varying intensities to control PTI and ETI responses. Therefore, pinpointing the role of ETI-related pathways is difficult because several authors have recently proposed interaction models that imply a connection between the two processes, where ETI improves PTI and vice versa, making it difficult to distinguish between the mechanisms and their specific gene players [72].

Other important genes that can be associated with an ETI response in BRS were the genotype-specific expressed clathrin assembly protein (Glyma.17G235500) and two TIR-NB-LRR resistance proteins (Glyma.01G046900 and Glyma.03G047900). A clathrin assembly protein was predicted to play an important role in *Arabidopsis* defense as an adenylate cyclase [73]. The role of nucleotide binding and a C-terminal leucine rich repeat domain (NLR) proteins carrying a Toll-interleukin 1 receptor (TIR) domain in the activation of lipase-like proteins, including EDS1 and SAG101, in plant immunity is well known [74]. Additionally, the TIR domain in these proteins signals cell death during plant defense [75,76].

### 3.3. A PIF7 Transcription Factor May Be Involved in the Regulation of Pathways during RLN Infection

Co-expressed, network, promoter, and genomic polymorphism analyses suggest Glyma.09G064200 may be a key gene in the RLN soybean response. In a further explanation of this hypothesis: (1) Glyma.09G064200 was differentially expressed in BRS (4 and 8 dpi) and TMG (2 dpi) but induced in both genotypes and all time-points (Appendix A). This gene encodes phyto-chrome-interacting factor (PIF7), which regulates sterol biosynthesis and plant thermosensitivity [77]. This TF may also regulate photosynthesis network genes and other important genes. (2) 18 DEGs are oppositely co-expressed in soybean and belong to a network in *Arabidopsis thaliana* and *Oriza sativa*, which are mostly photosynthesis-related (Figure 3a–d). (3) The promoter motif analysis identified a *bHLH* binding motif as a common *cis* element of these opposite co-expressed DEGs (Figure 3d). (4) Glyma.09G064200 presents a mutation at the amino acid Phe513Leu in BRS that is not present in TMG, having Willians 82 (W82) as reference genome, ditto that TMG and W82 are susceptible to RLN. Finally, due to the Glyma.09G064200 pattern of expression and the pathways described for this gene, the mutation may change pathway regulation under RLN infection, such as photosynthesis and sterol biosynthesis (phenylpropanoids), including jasmonate-responsive genes [78]. The *bHLH* transcription factor family, along with MYB (myeloblastosis related), NAC (no apical meristem (NAM), WRKY, and bZIP (basic leucine zipper), are involved in biotic and abiotic stress responses in plants [79]. A transcriptome of resistant chickpea infected with RLN, *Pratylenchus thornei*, identified regulation of several transcription factor families, especially 22 *bHLH* DEGs [80].

### 3.4. Intron Retention Event in BRS Transcript May Result in the Inactivation of Myo-Inositol Oxygenase

Alternative splicing can be directly affected by stresses and be associated with plant resistance [81,82,83,84], affecting the final proteome and regulating gene functions in plants, especially by events of intron retention and exon skipping [85,86,87]. Significant intron retention in BRS in the transcript encoding myo-inositol oxygenase (MIOX) (Glyma.05G224500) and intron splicing in TMG led us to hypothesize that truncating this gene in moderate resistance causes its loss of function and negatively affects nematode infection. All four *Arabidopsis* genes for myo-inositol oxygenase were expressed in syncytia induced by the beet cyst nematode *Heterodera schachtii* [88]. This study also showed that nematode susceptibility was significantly reduced in the quadruple myo-inositol oxygenase mutant.

## 4. Material and Method

### 4.1. Plant Material

The experiment used two G. max genotypes from Embrapa Soja Active Germplasm Bank (AGB). BRS and TMG were phenotypically classified as moderately resistant and high susceptibility to RLN infection by nematode reproduction factor standard method [22].

### 4.2. Nematode Infection Assays and Sample Preparation

For the phenotype assay (penetration activity and reproduction factor), seeds of each genotype were planted in a 5-gallon tray filled with sterile sand (Nematology Greenhouse—Embrapa Soja, Londrina, PR, Brazil). Five days after emergence, each healthy seedling was transplanted into a container filled with sterile sand. Three days after transplantion, each seedling was inoculated with 500 infectious forms of RLN (juveniles and adults), and control samples were not inoculated. Root samples were collected from infected and control plants, respectively, at 1/2, 1, 2, 4, 8, 17, and 35 days post inoculation (dpi). The selection of time-points range for samples collection was based on: (1) RLN lifecycle and behavior; (2) expected time of plant response to infection, based on PTI and ETI knowledge; (3) on our nematode penetration assay results; and (4) on previous literature with similar analyses [43].

For phenotyping, each biological replicate with three plants, was collected and individual nematodes were extracted with a 500-µm sieve and counted using a microscope and a 1 mL *Peters*’s slide. Comparisons of significant nematode penetration in roots were determined based on the statistical *T*-test, using SAS Statistics& Data Mining software v.7.1, SAS Institute Inc., Cary, NC, USA.

To capture transcript variation responses to *P. brachyurus* in *G. max*, we collected root samples from infected and control plants, respectively, at 1, 2, 4, and 8 dpi for RNA-seq run. Briefly, roots were excised, washed, flash frozen in liquid nitrogen, and stored at −80 °C until RNA extraction. Three plants for each biological replicate, and three biological replicates were collected for each genotype and time-point.

### 4.3. Library Construction, Sequencing, and Mapping of RNA-seq Reads

Total RNA was extracted from frozen samples using TRIzol^®^ reagents (Invitrogen), per manufacturer’s instructions. DNA contamination was removed using DNAse I (Invitrogen). Each biological repeat’s total RNA was pooled for library construction. The RNA TruSeq^TM^ SBS Sample Prep Kit v5-GA sample prep kit (Illumina, San Diego, CA, USA) was used to prepare the RNA-Seq library following the manufacturer’s instructions. The libraries were distributed into a flow cell for sequencing on an Illumina HiSeq 2000 for 101 bp length single-end reads with sequencing chemistry v4 (Illumina, San Diego, CA, USA). RNA-seq was performed by FASTERIS Biotechnology company (Geneva, Switzerland).

The initial base calling, adaptor trimming, and quality filtering of the reads generated with the Illumina analysis pipeline were performed using TrimGalore software [89] and checked by FastQC software [90]. High-quality mRNA-Seq reads were aligned to the *Glycine max* reference genome (*Glycine max* Wm82.a4.v1—Phytozome v.13) [91] using STAR RNA-seq aligner [92]. Duplicated reads were removed using PICARD v.2.23.

### 4.4. Differential Expression and Co-Expression Analysis

BAM files were read counted and processed by GFOLD to determine log2Fold change by comparing infected and non-infected at each time-point. GFOLD ranks differentially expressed genes from single biological replicate RNA-seq data, relying on the posterior distribution of log fold change, to overcomes the limitations of *p-value*, providing stable and biologically relevant results [93]. Differentially expressed genes (DEGs) were cut-off with “GFOLD (0.01)” values (≤−1 and ≥1) and log2FoldChange values (≤−2 and ≥2). DEGs were hierarchically clustered based on reads per kilobase of transcript, per million mapped reads (RPKM) and fold change values by events (time-points x genotypes) and treatments (inoculated/non-inoculated), applying Euclidean distance for similarity metric and complete linkage clustering method with software Cluster v3.0 and visualized with Java Treeview [94].

### 4.5. Genotype-Specific Expression Gene Analysis

Genotype-specific expression gene (GSEG) BRS and TMG gene expression to identify expression in only one genotype during infection. Transcripts per million (TPM) values were calculated per gene for all RNA-seq libraries using BAM and TPMcalculator [95]. The parameters for selecting GSEGs (genes) were TPM values ≥ 1.0 in all time-points of one of the genotype and TPM values = 0.0 in the other genotype. These data were used to identify GSEGs in BRS and TMG, in which an allele had consistent expression in all libraries of a genotype and complete absence in all libraries of the other.

### 4.6. Differential Splicing Events Analysis

New mapping files, with “EndToEnd” alignment parameter, were used to perform alternative splicing analyses with rMats v.4.0.2 software [96], by comparing the same genotype infected against non-infected in each correspondent time-point. The cut-off of significant differential splicing events (DSEs) was set to false discovery rate (FDR) values < 5% and IncLevel ≥ 0.2 in both compared treatments (inoculated and non-inoculated).

### 4.7. Variant Calling Based on RNA-seq Data

BAM files from all treatments were merged by genotype using Samtools, then N-in-CIGAR reads were split into multiple supplementary alignments and mismatching overhangs were clipped, according to the Genome Analysis Toolkit (GATK) v4.18. After variants were called using HaplotypeCaller and recalibrated, the identified SNP/InDel was filtered for minimal mapping depth (10), missing data (0.6), and minimal mapping quality (30) reads, using vcftools. Only sites identified on the genes as specific of the moderately resistant genotype (BRS), compared to the reference genome (Williams 82), and the variations found in TMG, were annotated for effect by SnpEff software.

### 4.8. Nematode Read Counting

After filtering and trimming step described above, reads were filtered out for possible contaminants, using bowtie2 mapping against rRNA from SILVA database, *Escherichia coli*, *Bradyrhizobium japonicum*, and *Glycine max.* The remaining reads were mapped against the *Pratylenchus penetrans* transcriptome (Nematode.net v.4 database), which presents 488 contigs. Total mapped reads were counted using Samtools v.1.8.

### 4.9. Gene Ontology (GO) and KEGG Analysis

A parametric analysis of gene set enrichment (PAGE) analysis was performed for the set of DEGs to detect significantly enriched or depleted GO categories, compared to the soybean genome (*Glycine max* Wm82.a2.v1) using the AgriGO tool v2.0 [97]. PAGE parameters were Fisher’s exact test, Bonferroni multi-test adjustment, the significance level was adjusted to *p*-value < 0.1, and the complete GO Slim database [97]. Ontology categories of genes presenting SNPs in BRS were classified based on the impact and GO Term Enrichment Tool-Soybase. KEGG analysis was performed using KOBAS [98], species *Glycine max* (soybean), protein sequence (first transcript of each gene) as input, and adjusted *p < 0.05* was considered statistically significant.

### 4.10. Network and Promoter Motif Analysis

After clustering analyses described in item 5.4., DEGs found co-expressed among time-points of genotypes but with inverse expression profile, (up-regulated in BRS and down-regulated in TMG at the same time-point, and vice versa), were selected for network and promoter motif analysis. In network analysis, the protein sequence of all 120 co-expressed genes was retrieved from Phytozome V13 and used as input on “STRING: functional protein association networks database and software for protein–protein interaction networks analysis” [99], selecting multiple sequences and *Glycine max* as organism. Sequences of 1 kb size upstream from the start codon of selected genes were retrieved from *Glycine max* Wm82.a4.v1 genome sequence based on coordinates using bedtools v2.29.2. A search for motif elements was performed with MEME software [100] and a comparison of identified motifs was performed with *Arabidopsis thaliana* DAP motifs [101] database using TOMTOM software [102].

### 4.11. Gene Expression Coefficient of Variation Analysis

TPM values from 16 libraries to measure gene expression levels. After calculating TPM values for each gene across all samples, the following filters were used to identify the most stable genes: (I) TPM values ≥ 0.1 in all 16 libraries; (II) average TPM ≥ 1.0; (III) coefficient of variation (CV = the ratio of the standard deviation (SD) to the arithmetic means (stdev/mean)) values lower than 0.1 [103]

### 4.12. RT-qPCR for Soybean Reference Genes and RNA-seq Data Validation

For quantitative RT-qPCR, samples were obtained from an independent assay carried out with the same parameters previously described in the sample preparation for RNA-seq. After total RNA extraction and DNAse I treatment, cDNA was synthesized with Super Script TM III Kit (Invitrogen) applying Oligo-dT primers, according to the manufacturer’s instructions and stored at −20 °C. Total RNA concentration was assessed using NanoDrop ND-1000 spectrophotometer ((Thermo Fisher Scientific Inc., Wilmington, NC, USA)).

Quantitative PCR was performed in 9.4 uL reaction volume and 384-well plate using the Maxima SYBR Green/ROX RT-qPCR Master Mix (2×) (Thermo Fisher Scientific Inc., Foster City, USA) on a detection system (ABI 7900HT [Life Technologies, Grand Island, New York, NY, USA]). The following thermal cycle settings were applied: 95 °C for 30 s, followed by 40 cycles of 95 °C for 5 s, and the final products obtained at 60 °C for 30 s. All reactions were repeated 3 times with 3 biological repeats; dissociation curves were checked to ensure the absence of any non-specific amplification. The efficiency of each primer pair used in this study was tested by amplifying a set of all the aforementioned samples to perform RT-qPCR reactions. All efficiencies tested were up to 90% (Appendix A).

Seven of the soybean’s most commonly used housekeeping genes: β-actin (Glyma.15G05570), elongation factor 1-alpha (Elf1-α) (Glyma.19G07240), elongation factor 1-beta (Elf1-β) (Glyma.02G44460), tubulin alpha (Tua) (Glyma.08G12140), and tubulin beta (Tub) (Glyma.20G27280) were tested by RT-qPCR analyses. BRS and TMG infected with RLN and non-infected samples were analyzed in three biological replicates with three technical replicates. Ct values of each reference gene candidate were analyzed using the RefFinder tool to generate a comprehensive ranking of the most stable genes [104].

To validate the RNAseq data, four DEGs were selected, and primers were designed using Primer3Plus (http://primer3plus.com/cgi-bin/dev/primer3plus.cgi: accessed on 1 March 2017). All primers were tested for amplification efficiency as described previously (Appendix A). The final relative quantification of each gene was calculated with REST software v.2.0.7 (2009), using *TUA* and *ELF1*-β as internal controls.

## 5. Conclusions

The soybean response to RLN infection was marked by a biphasic wave of DEGs with a peak of DEGs at a late stage (8 dpi). Early defense responses potentially related to PTI with oxidoreduction reactions, cell wall, and calcium binding were common across genotypes. A prominent response, distinct from the susceptible genotype, was initially observed in the moderately resistant genotype at 4 dpi. At this time-point, ETI related genes could be observed in BRS, including those of MAPK signaling, plant–pathogen interaction, and secondary metabolism pathways. RLN infection also causes an opposite expression profile of photosynthesis related genes between genotypes, showing downregulation in BRS, which together with the genomic polymorphism analysis revealed a key regulatory candidate for this condition, a transcription factor PIF7/ SREBP. Through analysis of genotype-specific expressed gene, we identified two BRS specific expressed genes, TIR-NB-LRR, which have been previously described to play a role in cell death signaling during resistance response. Additionally, the myo-inositol oxygenase gene identified in our alternative splicing analysis was shown to have a truncated alternative transcript in BRS, with an important role in blocking this pathway related to plant response against nematode and consequently redirecting the moderate resistance response. Our study provides new insights into plant–RLN interactions. The analysis of RLN-affected metabolic pathways and the identified genes are promising targets for further studies on their regulatory network and functional characterization in plant responses to RLN infection, with potential for plant breeding and the development of resistant materials.

## Figures and Tables

**Figure 1 plants-11-02983-f001:**
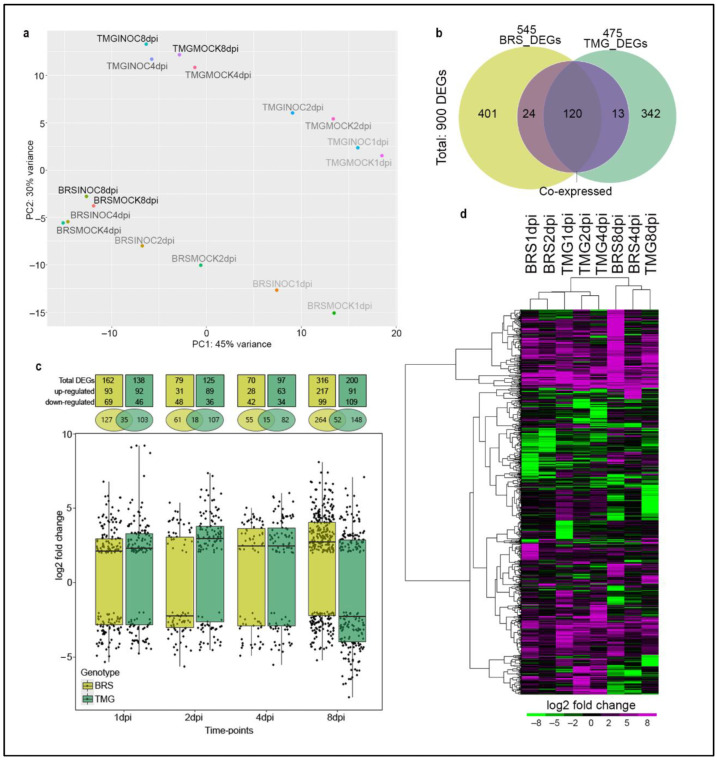
**Overview of soybean DEGs in moderate resistance and susceptible response to RLNs.** PCA of RNA-seq data with each color showing clear genotype, time-point, and treatment (infected and non-infected), where treatment is the differentiating factor among samples (**a**). Venn diagram showing unique and shared differentially expressed genes (DEGs) in BRS and TMG as common expressed genes under RLN infection, from left to right: (yellow) is DEGs exclusive to BRS = 401; (yellow + purple) is DEGs exclusive to BRS found in more than one time point = 24, (yellow + purple + green) is DEGs found in BRS and TMG, that could also have been expressed in more than one time point in each = 120; (green + purple) is DEGs exclusive to TMG found in more than one time point = 13; and (green) is DEGs exclusive to TMG = 342. (**b**). Boxplot with numbers of common expressed genes, up and down-regulated DEGs in BRS and TMG in response to RLN at 1, 2, 4, and 8 dpi (**c**). Hierarchical clustering of DEGs at each genotype/time-point (**d**).

**Figure 2 plants-11-02983-f002:**
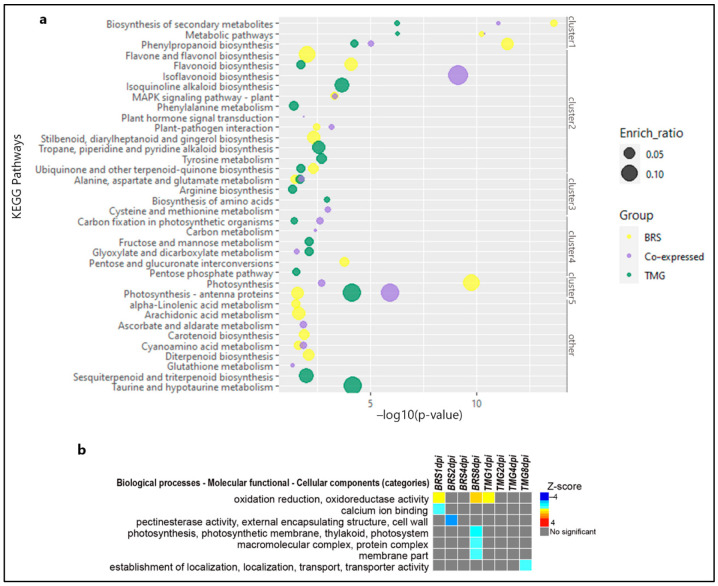
**General KEGG pathways and GO enrichment of DEGs.** Bubble plot of significant enriched KEGG pathways in BRS, TMG, and co-expressed DEGs under RLN infection. The enrich-ratio is represented by the size of each bubble as the pathway enrichment level, as well as their level of significance based on −log10(*p*-value) in the x-axis (**a**). PAGE results showing up- or down-regulated gene ontology categories in soybean moderately resistant and susceptible genotypes during the four time-points of RLN infection. The statistical significance is inferred using a Z-score value. As a two-tailed test, the Z-score can be either positive or negative. Either the term has a positive Z-score and the fold-change mean of all associated genes is upregulated, or the term has a negative Z-score and the fold-change mean is downregulated [36] (**b**).

**Figure 3 plants-11-02983-f003:**
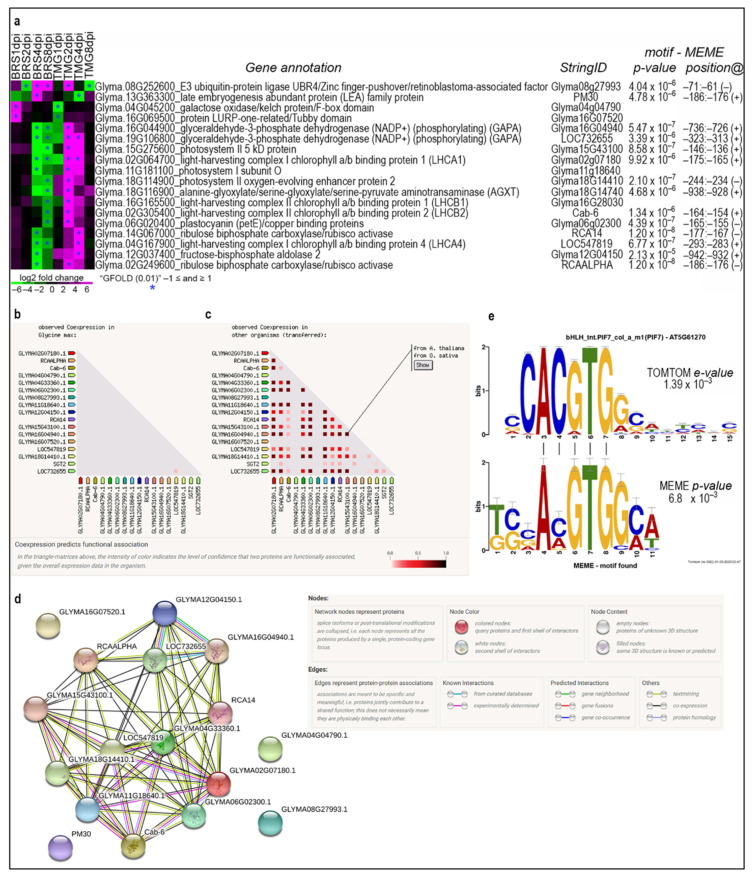
**Co-expressed genes showing inverted profiles between genotypes.** Heatmap showing the 18 DEGs identified as co-expressed with inverted profiles between genotypes (**a**). String co-expression data showing that these genes were not previously found as co-expressed in soybean (**b**). String co-expression data showing that these genes are highly co-regulated in other plant species (**c**). The expression network showing the predicted interaction of 13 of the 18 soybean genes showed significantly inverted expression profiles between genotypes, and genes interconnected with purple line means experimentally determined known interaction (**d**). Alignment of the promoter motif identified in these 18 genes by MEME with the motif of a plant non-canonical *E-box* cis element by TOMTOM (**e**). Figure 3a shows correspondence of current and former soybean gene model IDs in Figure 3b–d from the String Database.

**Figure 4 plants-11-02983-f004:**
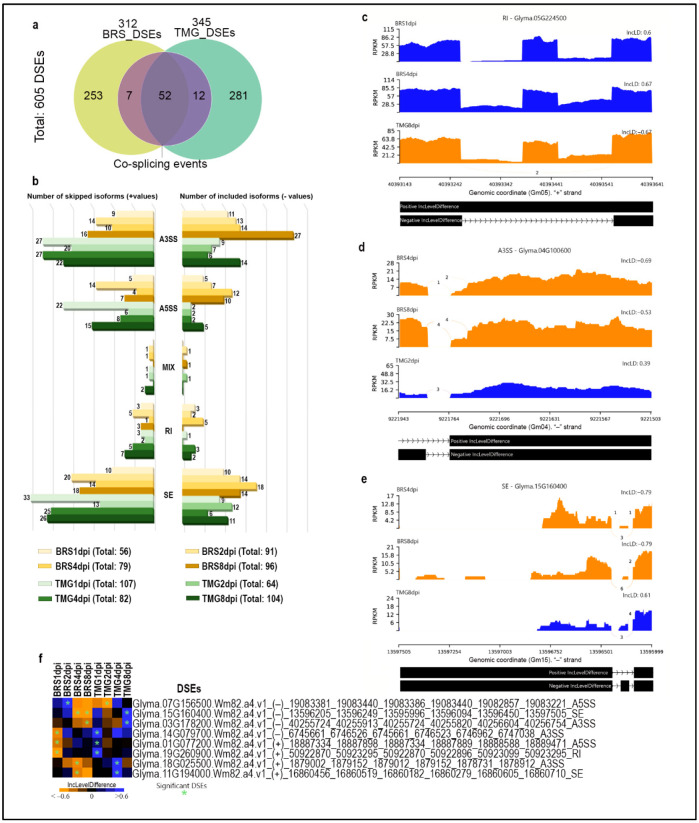
**Soybean alternative splicing events landscapes in response to RLN.** Venn diagram showing unique and shared differentially splicing events (DSEs) in BRS and TMG under RLN infection (**a**). Numbers of skipping and inclusion isoforms of splicing events in BRS and TMG in response to RLN at 1, 2, 4, and 8 dpi (**b**). Sashimi-plot of differential splicing of retained intron (**c**), 3′untranslated regions (**d**), and skipped exons(**e**). Heatmap showing the impact of RLN infection on 8 DSEs that presented inverted profiles between genotypes (**f**).

**Figure 5 plants-11-02983-f005:**
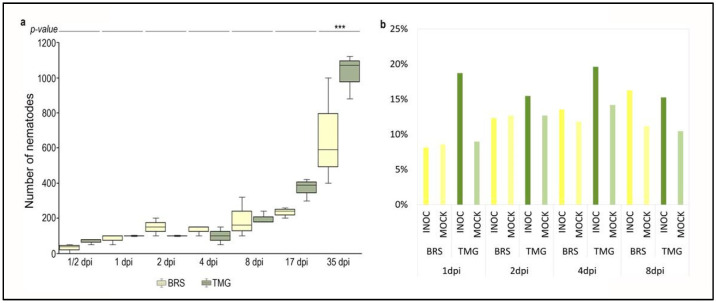
**Nematode reads in RNA-seq data and infection progress.** Boxplot of nematode counting from BRS and TMG roots during the course of RLN infection (**a**). *** Significant difference between BRS and TMG with *p-value* < 1%. Percentage of mapped reads against *Pratylenchus penetrans* transcriptome (mapped reads in mock samples is due to rRNA genes) (**b**).

**Table 1 plants-11-02983-t001:** Genotype-specific expressed genes (GSEGs) identified in BRS and TMG and respective functional annotation.

	Gene_ID	Annotation
**TMG-GSEGs**	Glyma.04G195633	
	Glyma.06G241300 *	ankyrin repeat family protein-related
	Glyma.06G241600 *	ankyrin repeat family protein-related
	Glyma.06G242000 *	ankyrin repeat family protein-related
	Glyma.07G094051	
	Glyma.12G110550	
	Glyma.14G136300	Phytochromobilin synthase/Phytochromobilin:ferredoxin oxidoreductase
	Glyma.U031724	
	Glyma.18G208300	UDP-glucosyl transferase/Soyasapogenol B glucuronidegalactosyltransferase
	Glyma.20G076400	
**BRS-GSEGs**	Glyma.01G046900	Toll—interleukin 1—resistance/leucine-rich repeat-containing protein/NB-ARC domain (LRR)
	Glyma.02G089500	
	Glyma.02G090200	Zinc-finger of C2H2 type
	Glyma.03G047900	Toll—interleukin 1—resistance/leucine-rich repeat-containing protein/NB-ARC domain (LRR)
	Glyma.U033005	
	Glyma.04G115300	UDP-Glycosyltransferase/glycogenphosphorylase
	Glyma.04G132300	
	Glyma.08G151300	
	Glyma.14G019500	Succinate-semialdehydedehydrogenase (NAD(+))
	Glyma.15G240300	
	Glyma.16G078600	
	Glyma.16G112400	
	Glyma.17G235500	clathrin assembly protein/ANTH domain
	Glyma.19G070966	

* In tandem genomic location.

## Data Availability

Raw data is available on NCBI’s Short Read Archive (SRA) database under accession number SRR11523763-78.

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
