# Peer review of "Time Course RNA-seq Reveals Soybean Responses against Root-Lesion Nematode and Resistance Players"

_plants, 2022, doi:10.3390/plants11212983_

Round 1

Reviewer 1 Report

The paper of Lopes-Caitar et al describes the transcriptome profile of different soybean genotypes against the root lesion nematode Pratylenchus brachyurus. The presented data is interesting in the light of the plant responses to this particular nematode, using time course RNA-seq data. The paper shows relevant information and is interesting for scientists working in molecular research of plant-parasitic nematodes. Although the paper provides relevant information, some of the analyses are either non informative, could be done in more precise manner or might be considered preliminary due to the lack of effective mRNA-seq biological replicates. Depending on the policy of your journal, this study can be considered acceptable, however, with the caveat of the low number of technical mRNA-seq replicates.

Major remarks

Lines 669-673 and 677 and 678: the RNA collected from the different biological replicates were pooled together for RNA-seq, which in fact will constitute an overall single biological replicate. Meaning that all subsequent analyses are based on one biological mRNA-seq replication for each time point after nematode infection. For such type of experiments, a minimum of three effective sequenced bio-replicates are required to produce accurate results. The overall quality of the RNA-seq data often depends on the reproducibility among replicates. Therefore, appropriate number of biological replicates is of critical importance because low replicate number have insufficient statistical power to correctly predict DEGs. Besides, replicates are also essential to resolve false-positive calls such as mapping artifacts or possible contaminations. Therefore, analyzing RNA-seq data without effective sequenced biological replicates might have a more preliminary rather than conclusive value.

Lines 253-255: Not very clear why the unmapped reads where mapped to P. penetrans transcriptome, especially to a very incomplete dataset (i.e., 488 contigs only). Would it possible to generate a de novo assembly of the non-mapped reads against soybean, and then remapped those unmapped reads to the new transcriptome? This would not only provide a more accurate result but could also provide additional information of the nematode DGEs in both soybean cultivars. This could corroborate the hypothesis discussed in lines 457-471.

Other remarks:

Line 54: Not clear the meaning that “Plant-RLN interaction are considered new in terms of evolution”. In fact, should be the opposite, the interaction of sedentary nematodes is considered more specialized and newer in the light of plant-nematode parasitism evolution.

Line 94-111: The authors should reference other recent studies on this field such as: Channale et al. 2021 (Scientific Reports 11: 17491); Rahaman et al. 2021 (Plant Molecular Biology 106: 381-406); Vieira et al. 2019 (Front Plant Sci 10: 971).

Lines 174-255: Instead of using the gene codes “Glyma.0…”, I would suggest including the gene function instead. This will provide a much better understanding of DGEs among both cultivars tested.

Lines 658: Can the authors clarify what are infectious forms of RLN? 

Author Response

Thank you for taking the time and read our manuscript. All your suggestions are very important to us. Please, find more in our cover letter for you.

Reviewer 2 Report

The paper by Lopes-Caita et al., is very intersting with a lot of information on gene expression differences between the two genotypes of soybean. Thus I believe that it must be published because this is the first time that authors describe what happens in plant in presence of RLN. My only concern is on the length of the paper and a lot of expression data  described. I suggest to reduce and to be shorter even in the discussion section.

Author Response

(The authors gave the same response as above.)

Round 2

Reviewer 1 Report

The authors have follow some recommendations to improve the quality of the manuscript. The authors also acknowledge that performing additional biological replicates will be ideal for such type of work. In my opinion, this is one of the weakest points of the manuscript and will have an overall impact on the downstream analyses and interpretation of the results.

As mentioned before, I will leave the final decision to the editors of the journal to decide whether the use of a single technical replication will be satisfactory for such type of work.